# The Role of Cytokine-Inducible SH2 Domain-Containing Protein (CISH) in the Regulation of Basal and Cytokine-Mediated Myelopoiesis

**DOI:** 10.3390/ijms241612757

**Published:** 2023-08-14

**Authors:** Wasan Naser, Saeed Maymand, Daniel Dlugolenski, Faiza Basheer, Alister C. Ward

**Affiliations:** 1School of Medicine, Deakin University, Geelong, VIC 3216, Australia; wasan.aboud@sc.uobaghdad.edu.iq (W.N.); saeed.maymand@education.vic.gov.au (S.M.); daniel.dlugolenski@10xgenomics.com (D.D.); faiza.basheer@deakin.edu.au (F.B.); 2College of Science, University of Baghdad, Baghdad 10071, Iraq; 3Institute for Mental and Physical Health and Clinical Translation, Deakin University, Geelong, VIC 3216, Australia

**Keywords:** CISH, cytokine, G-CSF, GM-CSF, granulopoiesis, myelopoiesis, neutrophil, SOCS

## Abstract

Cytokine-inducible SH2 domain-containing protein (CISH) is a member of the suppressor of cytokine signaling (SOCS) family of negative feedback regulators shown to play crucial roles in lymphoid cell development and function as well as appetite regulation. It has also been implicated in the control of signaling downstream of the receptors for the cytokines granulocyte/macrophage colony-stimulating factor (GM-CSF) and granulocyte colony-stimulating factor (G-CSF) in myeloid cells. To investigate the physiological role of CISH in myelopoiesis, mice deficient in CISH were analyzed basally and in response to administration of these cytokines. CISH knockout (KO) mice possessed basally elevated neutrophils in the blood, bone marrow, and spleen compared to wild-type (WT) mice. During GM-CSF-induced myelopoiesis, the frequency of neutrophils, myeloid dendritic cells (DCs), and CFU-M in the bone marrow was higher in the KO, as were the neutrophils and CFU-G in the spleen. In contrast, no differences were observed between KO and WT mice during G-CSF-induced myelopoiesis apart from an elevated frequency of CFU-G and CFU-M in the spleen. This work has identified a role for CISH in the negative regulation of granulopoiesis, including that mediated by GM-CSF.

## 1. Introduction

The suppressor of the cytokine signaling (SOCS) family of proteins function as critical negative feedback regulators, enabling the timely termination of cytokine signaling as part of normal development and homeostasis [1,2]. The cytokine-inducible SH2-containing (CISH) protein was the founding SOCS family member [3]. It has been shown to be induced by numerous cytokines implicated in blood and immune development, including erythropoietin (EPO), granulocyte colony-stimulating factor (G-CSF), granulocyte/macrophage (GM)-CSF, and interleukin (IL)-2, IL-3, IL-4, and IL-15 [3,4,5], but also after T cell receptor (TCR) stimulation [6]. 

Previous studies have demonstrated that CISH plays a number of key roles in lymphoid cells. For instance, transgenic mice expressing CISH from the β-actin promoter showed altered T and NK cell responses due to partial suppression of IL-2R signaling [7]. CISH knockout (KO) mice showed preferential T cell differentiation into Th2 and Th9 cells [4] and Treg cell differentiation into Th2 cells [8] due to the dysregulation of IL-4R signaling, resulting in the spontaneous development of allergic pulmonary disease with excessive eosinophil influx [4,8]. CISH deletion in NK cells resulted in an increase in NK cell proliferation, differentiation, and cytotoxicity against tumors due to enhanced IL-15R signaling [5]. CISH ablation also resulted in altered CD8+ T cell functionality due to dysregulated TCR signaling [9]. 

However, our understanding of the physiological role of CISH in the context of myelopoiesis remains incomplete. Studies ex vivo have demonstrated that GM-CSF can substantially induce CISH expression [3,10] and also implicated CISH in the negative regulation of GM-CSF-mediated bone marrow-derived dendritic cell (BMDC) development [11]. Additional in vivo work has revealed a key role for CISH in limiting GM-CSF receptor signaling in inflammatory states [10]. Other in vitro studies have shown that G-CSF can also induce modest CISH expression, with CISH able to interact indirectly with the G-CSF receptor, suggesting that CISH may regulate G-CSFR signaling [12]. To further explore the regulation of myelopoiesis by CISH a recently described *Cish* KO mouse line [13] was analyzed with respect to basal and cytokine-induced myelopoiesis.

## 2. Results

### 2.1. Role of CISH in Basal Myelopoiesis

To investigate a potential role for CISH in basal myelopoiesis, the blood, bone marrow, and spleen of 11-week-old Balb/c *Cish*^+/+^ (wild-type, WT) and *Cish*^−/−^ (KO) mice [13] were examined. Differential counts revealed a significant increase in total neutrophils in the peripheral blood of *Cish*^−/−^ compared to *Cish*^+/+^ mice (Figure 1A), but no other changes were observed, including in the bone marrow (Figure 1B) and spleen (Figure 1E). However, analysis of specific myeloid populations by FACS [14] identified an increased frequency of neutrophils and total myeloid cells in *Cish*^−/−^ mice in both the bone marrow (Figure 1C) and spleen (Figure 1F). Colony-forming assays showed no statistically significant differences in the bone marrow (Figure 1D) or spleen (Figure 1E) between genotypes, nor were changes observed in the overall cellularity of the bone marrow (WT: 4.36 ± 0.25 × 10^7^; KO: 4.41 ± 0.27 × 10^7^; *p* = 0.882) or spleen (WT: 6.21 ± 0.66 × 10^7^; KO: 6.69 ± 0.71 × 10^7^; *p* = 0.625).

### 2.2. Role of CISH in GM-CSF-Induced Myelopoiesis

To further explore the in vivo role of CISH in GM-CSF-induced myelopoiesis, *Cish*^+/+^ and *Cish*^−/−^ mice were injected with GM-CSF, and the blood, bone marrow, and spleen analyzed. Compared to control injection with vehicle (PBS), GM-CSF injection resulted in a significant increase in total neutrophils in the peripheral blood only in *Cish*^−/−^ mice, although lymphocytes were decreased in both *Cish*^+/+^ and *Cish*^−/−^ mice (Figure 2A,B). However, no differences were observed between GM-CSF-treated *Cish*^+/+^ and *Cish*^−/−^ mice, except in the percentage of lymphocytes, which were significantly lower in the *Cish*^−/−^ cohort (Figure 2A,B).

In the bone marrow, GM-CSF injection caused a significant increase in total and mature neutrophils and a reduction in lymphocytes and normoblasts in both *Cish*^+/+^ and *Cish*^−/−^ mice, and band cells just in *Cish*^+/+^ mice, although there were no differences between genotypes with either vehicle or GM-CSF treatment (Figure 2C,D). FACS analysis revealed a significant increase in the frequency of neutrophils and myeloid DCs, along with total DCs and total myeloid cells, in both genotypes, but they were significantly higher in each case in GM-CSF-treated *Cish*^−/−^ mice compared to similarly treated *Cish*^+/+^ mice (Figure 2E). Colony-forming assays revealed that GM-CSF administration resulted in a significant increase in the frequency of CFU-G and CFU-M in *Cish*^−/−^ mice only, with the frequency of CFU-M in GM-CSF-treated *Cish*^−/−^ mice elevated in comparison to *Cish*^+/+^ mice treated in the same manner (Figure 2F). There was no statistically significant change in total cellularity in either *Cish*^+/+^ or *Cish*^−/−^ mice following GM-CSF injection (Figure 2G).

In the spleen, GM-CSF injection caused significant increases in the frequency of total and mature neutrophil populations in both *Cish*^+/+^ and *Cish*^−/−^ mice, with a significant reduction in lymphocytes and normoblasts (Figure 2H,I). However, no significant differences were observed between genotypes in response to either vehicle or GM-CSF treatment (Figure 2H,I). FACS analysis showed that GM-CSF injection caused statistically significant increases in the frequency of myeloid DCs and total DCs in both genotypes, with a significant increase in total myeloid cells in *Cish*^+/+^ mice only, but again, no significant differences between genotypes were seen (Figure 2J). Colony-forming assays revealed no significant changes in *Cish*^+/+^ mice in response to GM-CSF administration, but a statistically significant increase in the frequency of CFU-G in GM-CSF-treated *Cish*^−/−^ mice was observed that was also significantly higher compared to similarly treated *Cish*^+/+^ mice (Figure 2K). In the spleen, GM-CSF injection caused an increase in splenic cellularity in both *Cish*^+/+^ and *Cish*^−/−^ mice, although this reached statistical significance only in *Cish*^−/−^ mice (Figure 2L). This resulted in the total number of spleen neutrophils being significantly elevated in response to GM-CSF in *Cish*^−/−^ compared to *Cish*^+/+^ mice (WT: 1.44 ± 0.06 × 10^6^; KO: 2.09 ± 0.17 × 10^6^; *p* = 0.037).

### 2.3. Role of CISH in G-CSF-Induced Myelopoiesis

To directly investigate the potential in vivo role for CISH in G-CSF-mediated myelopoiesis, *Cish*^+/+^ and *Cish*^−/−^ mice were injected with G-CSF, and the blood, bone marrow, and spleen analyzed. G-CSF injection resulted in an obvious and statistically significant increase in blood neutrophils—total and mature—in both *Cish*^+/+^ and *Cish*^−/−^ mice, with a concomitant decrease in lymphocytes (Figure 3A,B). However, G-CSF treatment failed to elicit significant differences between *Cish*^+/+^ and *Cish*^−/−^ mice (Figure 3A,B).

In the bone marrow, G-CSF injection resulted in significant increases in the percentage of total neutrophils, metamyelocytes, and mature neutrophils as well as decreases in lymphocytes and normoblasts for both genotypes, with band cells increasing just in *Cish*^−/−^ mice (Figure 3C,D). However, no significant differences between *Cish*^+/+^ and *Cish*^−/−^ mice were observed (Figure 3C,D). FACS analysis demonstrated a significant increase in the frequency of macrophages, neutrophils, and total myeloid cells in both *Cish*^+/+^ and *Cish*^−/−^ mice following G-CSF treatment, but no statistically significant differences were observed between genotypes apart from an increased frequency of myeloid cells and neutrophils in vehicle-treated *Cish*^−/−^ compared to *Cish*^+/+^ mice (Figure 3E). Colony-forming assays failed to reveal significant changes between G-CSF-treated and vehicle-treated mice of either genotype, nor between *Cish*^+/+^ and *Cish*^−/−^ mice, with either treatment (Figure 3F). G-CSF injection resulted in a reduction in bone marrow cellularity in both *Cish*^+/+^ and *Cish*^−/−^ mice, although this only reached statistical significance in *Cish*^−/−^ mice (Figure 3G).

In the spleen, G-CSF treatment caused increases across most neutrophil populations of both genotypes, reaching significance for total neutrophils, promyeloblasts, and mature neutrophils, with lymphocytes significantly decreasing, but again no differences between genotypes were seen (Figure 3H,I). FACS analysis further demonstrated that G-CSF injection resulted in a significant increase in the frequency of neutrophils and total myeloid cells in both *Cish*^+/+^ and *Cish*^−/−^ mice, although there were no differences between genotypes (Figure 3J). Colony-forming assays also revealed a statistically significant increase in the frequency of CFU-G, CFU-M, and CFU-GM following G-CSF treatment in both *Cish*^+/+^ and *Cish*^−/−^ mice, with the frequency of CFU-G and CFU-M significantly elevated in G-CSF-treated *Cish*^−/−^ compared to similarly treated *Cish*^+/+^ mice (Figure 3K). G-CSF treatment caused a statistically significant increase in splenic cellularity in both *Cish*^+/+^ and *Cish*^−/−^ mice, but there was no difference between genotypes (Figure 3L).

## 3. Discussion

CISH has been identified as a physiological regulator of immune cell development and function [2], particularly through its actions on the IL-4R [4] and T cell receptor [9] in specific T cell populations, the IL-15R in NK cells [5], and the GM-CSFR in DCs and other myeloid cells [10,11], with in vitro studies suggesting a potential role in regulating G-CSFR [12]. This study used a recently described *Cish*-deficient mouse line [13] to investigate the impact of CISH on myelopoiesis. The results indicate that CISH contributes to the regulation of both basal and cytokine-induced myelopoiesis mediated by GM-CSF, principally impacting neutrophil numbers (Table 1).

An analysis of basal myelopoiesis revealed small but significant differences in *Cish*^−/−^ compared to *Cish*^+/+^ mice. There was a significant increase in total circulating neutrophils, with an increased frequency of CD11b^+^ CD11c^−^ Ly6G^+^ neutrophils—and, indeed, total myeloid cells—in the bone marrow and spleen of *Cish*^−/−^ mice (Figure 1). Not all of these parameters reached significance in the control mice used in the GM-CSF (Figure 2) and G-CSF (Figure 3) studies, but the majority showed a similar trend, with the smaller numbers likely impacting the ability to demonstrate statistical significance. An analysis of mice on a C57/BL6 background [13] also confirmed a significant increase in bone marrow myeloid cells (WT: 10.14 ± 1.18%; KO: 13.90 ± 1.01%; *p* = 0.037) and neutrophils (WT: 7.37 ± 0.93%; KO: 10.34 ± 0.73%; *p* = 0.031). Overall, these results suggest a non-redundant role for CISH in the regulation of basal neutrophil levels. Amongst other SOCS proteins, SOCS3 has also been found to have a role in basal granulopoiesis, with hematopoietic-specific SOCS3 deletion resulting in neutrophilia and splenomegaly, although only in more mature mice, with this attributed to enhanced G-CSF signaling [15].

This led us to investigate potential cytokines that may have their myelopoietic actions regulated by CISH. GM-CSF represents a key regulator of myelopoiesis during inflammation as well as during DC development [16], with CISH implicated as a negative regulator of GM-CSFR signaling in the context of inflammation [10] and during GM-CSF-mediated bone marrow-derived dendritic cell (BMDC) development [11]. Therefore, *Cish*^−/−^ and *Cish*^+/+^ mice were compared with respect to GM-CSF-mediated myelopoiesis. Following GM-CSF treatment, both *Cish*^−/−^ and *Cish*^+/+^ mice exhibited a large increase in mature and total neutrophil populations as well as CD11b^+^ CD11c^+^ MHCII^+^ myeloid DCs in the bone marrow and spleen, which is consistent with the results of other studies examining GM-CSF administration in mice [17,18]. GM-CSF treatment also caused a statistically significant increase in total blood neutrophils, CFU-G and CFU-M in the bone marrow, and CFU-G and overall cellularity in the spleen only in *Cish*^−/−^ mice, with the frequency of bone marrow neutrophils, myeloid DCs, and CFU-M and spleen CFU-G significantly higher than in similarly treated *Cish*^+/+^ mice. However, because of the enhanced cellularity, the total number of splenic neutrophils in *Cish*^−/−^ mice also significantly exceeded those in *Cish*^+/+^ mice. Collectively, this suggests that CISH plays a negative regulatory role in the GM-CSF-mediated production of both myeloid DCs and neutrophils in the bone marrow and spleen. The former is consistent with a previous study showing that CISH knockdown led to excessive GM-CSF-mediated DC production from isolated bone marrow cells [11]. Meanwhile, others have shown that the absence of CISH results in an enhanced responsiveness of neutrophils to GM-CSF [10]. These authors showed that *Cish* mRNA was strongly induced by GM-CSF, with CISH able to directly interact with the GM-CSFR beta-chain to regulate cell surface receptor levels, thereby impacting the length of activation of the associated JAK2 and the downstream STAT5 transcription factor, providing clear mechanistic details.

G-CSF is a critical regulator in both basal and emergency granulopoiesis [19,20], with G-CSF demonstrated to induce CISH expression and a suggestion that it may regulate G-CSFR signaling [12]. Therefore, the potential role of CISH in G-CSF-mediated myelopoiesis was also investigated. Following G-CSF treatment, *Cish*^+/+^ mice exhibited a dramatic increase in multiple neutrophil populations in the blood, bone marrow, and spleen, with the difference most evident in the latter, where the frequency of these and various CFU populations were markedly increased in concert with a substantial enhancement in overall cellularity, reflected in the overall size of the spleen (WT+PBS: 0.100 ± 0.05 g; WT+G-CSF: 0.283 ± 0.10 g; *p* = 0.000074), which is consistent with previous studies [21]. *Cish*^−/−^ mice showed similar significant increases in the same populations following G-CSF treatment, with only CFU-G and CFU-M frequencies in the spleen significantly different from those in *Cish*^+/+^ mice. An analysis of G-CSF-mediated activation of STAT3 and STAT5 in bone marrow cells revealed no difference in extent or duration between *Cish*^+/+^ and *Cish*^−/−^ mice (Appendix A). These results suggest that CISH does not exert a major redundant negative regulatory role in G-CSFR signaling, including that involved in G-CSF-mediated neutrophil production. This is in stark contrast with SOCS3, with hematopoietic-specific *Socs3* KO mice exhibiting a prolonged and excessive response to G-CSF injection, leading to a significant increase in cells along the neutrophil lineage [15]. This difference may relate to the G-CSF induction of CISH being less than that of SOCS3—and, indeed, much less than the GM-CSF induction of CISH [10]—as well as the inability of CISH to directly dock to the G-CSFR [12].

CISH has been implicated in the susceptibility of humans to a variety of infectious diseases [22,23]. Therefore, it remains intriguing as to how the differences in innate immunity in *Cish*^−/−^ mice described here might contribute to this. Studies exploring viral and parasitic infections in *Cish*^−/−^ mice are currently underway that aim to directly address the impacts of CISH on the pathogenesis of these agents.

## 4. Materials and Methods

### 4.1. Animal Husbandry

This study used previously described *Cish*^+/+^ wild-type (WT) and *Cish*^−/−^ knock-out (KO) mice, principally those on a Balb/c background but also some C57/BL6 [13]. Mice were maintained on a standard rodent chow diet and experienced a 12-hour light/dark cycle, with their genotype determined as described [13]. All animal work was carried out with the approval of the Deakin University Animal Ethics Committee, which is subject to the Australian Code for the Responsible Conduct of Research. ARRIVE 2.0 guidelines were followed throughout. The sample size was based on preliminary experiments, with no blinding or randomization performed. Mice showing signs of illness were excluded from the study. Cytokine administration involved intraperitoneal injection of 10-week-old female mice with 6 µg/kg recombinant human G-CSF (rhG-CSF) (Ristempa, Pegfilgrastim) in 150 µL sterile phosphate buffered saline (PBS) that exhibits equivalent binding as its murine counterpart [24] and 30 µg/kg recombinant mouse GM-CSF (rmGM-CSF) (Cat# 415-ML-050, R&D Systems, In Vitro Technologies, Noble Park North, Australia) in 100 µL PBS – or equivalent volumes of PBS alone – daily for 5 consecutive days (basal) with humane culling 1 day later for tissue collection.

### 4.2. Tissue Collection and Histochemistry

Blood was collected from the tail prior to euthanasia by cervical dislocation, with single-cell suspensions generated from aspirated bone marrow or dissected spleen and passed through a 40 µm cell strainer. An aliquot of each cell suspension was mixed with trypan blue, and vital cell counts were performed with a Countess™ Automated Cell Counter (Invitrogen Australia Pty. Ltd., Mount Waverley, Australia). Smears were prepared on poly-L-lysine-coated glass slides (ProSciTech, Kirwan, Australia) using a cytospin funnel (Thermo Fisher Scientific Australia Pty. Ltd., Scoresby, Australia) and fixed in absolute methanol for 1 min prior to staining with 10% (*v*/*v*) Giemsa (Sigma-Aldrich Pty. Ltd., Castle Hill, Australia) for 20 min. Slides were examined on a Leica DME stereomicroscope and differential counts performed, with images captured on a Leica DFC290 digital camera controlled by the Leica Application Suite for Windows (Leica Microsystems, Macquarie Park, Australia).

### 4.3. FACS Analysis

Approximately 2 × 10^6^ cells from the bone marrow or spleen were resuspended in 2% (*v*/*v*) fetal bovine serum and 0.1% (*w*/*v*) EDTA in PBS and incubated with Fc block solution (anti-CD16/CD32) (BD Bioscience, North Ride Australia) before analysis with an antibody cocktail containing PI-PerCP-Cy5-5, anti-CD45.2-FITC (#104), anti-CD11c-APC (#HL3), anti-CD11b-PE (#M1/70), anti-Ly6C-PE-Cy7 (#1A8), anti-Ly6G-BV421 (#AL-21), and anti-MHCII(I-A/I-E)-BV480 (#M5/114.15.2) (BD Bioscience, Thermo Fisher Scientific) to quantify specific myeloid populations, using a previously described protocol [14]. A minimum of 100,000 viable events were recorded for each sample from independent experiments with a BD FACS-Canto II flow cytometer and analyzed using BD FACSDiva software (v6.0) (Appendix A).

### 4.4. Colony-Forming Assays

A total of 5 × 10^5^ cells from the bone marrow or spleen were added to 5 mL of methylcellulose media (R&D System), and 1 mL was poured into a 35 mm dish (Thermo Fisher Scientific). These were incubated in a humid atmosphere at 37°C with 5% CO2, and the number of CFU-G, CFU-M, and CFU-GM was enumerated on day 14.

### 4.5. Electropheoretic Mobility Shift Assays

Isolated bone marrow cells were placed in a tissue culture flask for 1 h, with the non-adherent cells transferred to a fresh flask for 3 h before stimulation of 1 × 10^6^ cells with rhG-CSF for up to 120 min, following which nuclear extracts were prepared and electrophoretic mobility shift assays (EMSA) performed to analyze activation of STAT3 and STAT5, as described [25].

### 4.6. Statistical Analysis

Physiological data were analyzed with GraphPad Prism 8.0 using a two-way analysis of variance (ANOVA)/Tukey’s multiple comparison test, or multiple *t*-tests using the Bonferroni–Dunn method, with *p* < 0.05 considered significant in all cases.

## Figures and Tables

**Figure 1 ijms-24-12757-f001:**
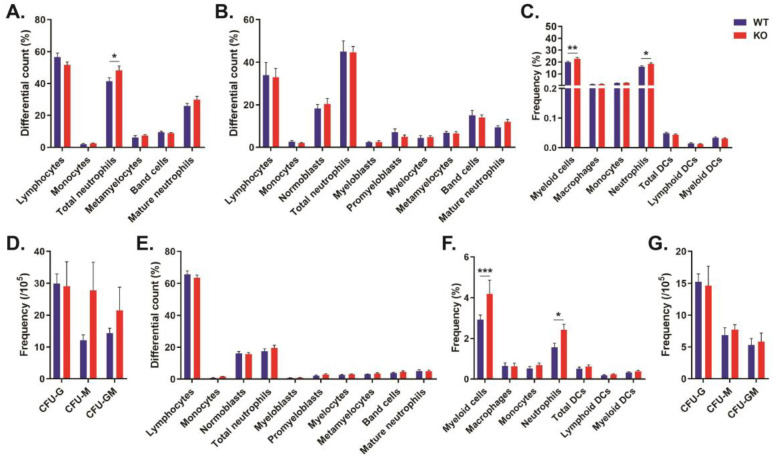
Effect of CISH ablation on basal white blood cells. Analysis of blood (**A**), bone marrow (**B**–**D**), and spleen (**E**–**G**) from *Cish*^+/+^ (WT) and *Cish*^−/−^ (KO) mice as indicated, presenting differential counts of Giemsa-stained smears (**A**,**B**,**E**), frequencies of specific myeloid populations determined using FACS (**C**,**F**), and frequencies of relevant colonogenic populations (**D**,**G**). Shown are the mean and standard error of the mean (SEM), along with statistical significance as determined by multiple *t*-tests between genotypes using the Bonferroni–Dunn method (* *p* < 0.05, ** *p* < 0.01, *** *p* < 0.001; n = 9).

**Figure 2 ijms-24-12757-f002:**
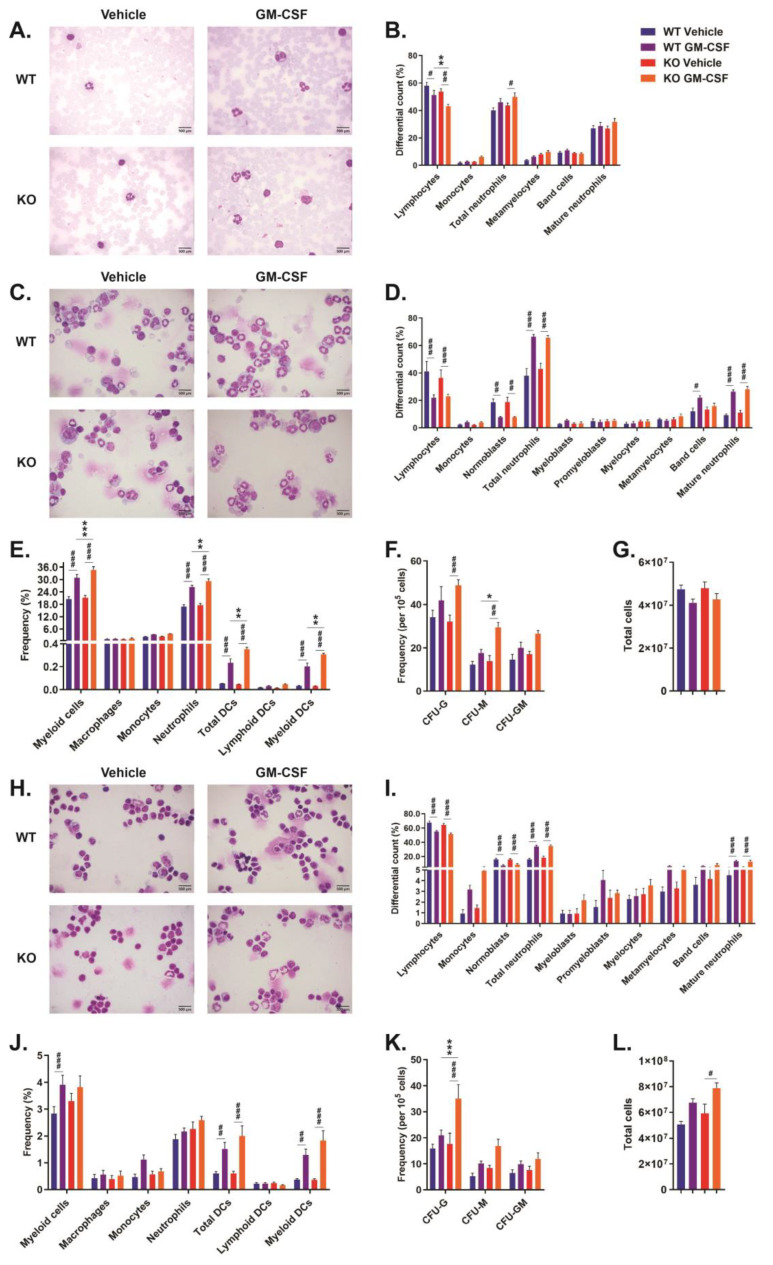
Effect of CISH ablation on GM-CSF-mediated myelopoiesis. Analysis of blood (**A**,**B**), bone marrow (**C**–**G**), and spleen (**H**–**L**) from *Cish*^+/+^ (WT) and *Cish*^−/−^ (KO) mice injected daily with either PBS (Vehicle) or recombinant mouse granulocyte/macrophage colony-stimulating factor (GM-CSF) as indicated and sacrificed on day 6. Shown are representative images of Giemsa-stained smears (**A**,**C**,**H**) along with corresponding differential counts (**B**,**D**,**I**), frequencies of specific myeloid populations determined using FACS (**E**,**J**), frequencies of relevant clonogenic populations (**F**,**K**), and total cell counts (**G**,**L**). Shown are the mean and SEM, together with statistical significance as determined by two-way analysis of variance (ANOVA)/Tukey’s multiple comparison test indicated between genotypes (* *p* < 0.05, ** *p* < 0.01, *** *p* < 0.001) or treatments (# *p* < 0.05, ## *p* < 0.01, ### *p* < 0.001) (n = 6).

**Figure 3 ijms-24-12757-f003:**
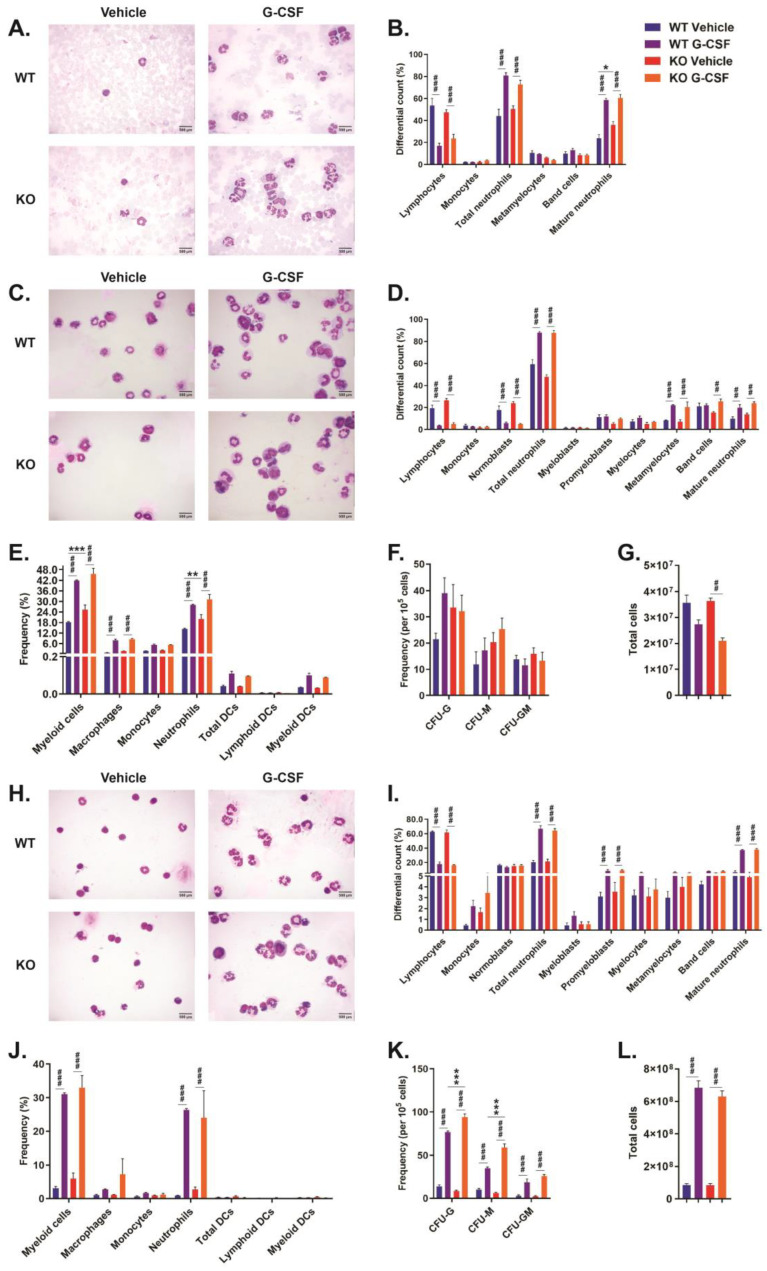
Effect of CISH ablation on G-CSF-mediated myelopoiesis. Analysis of blood (**A**,**B**), bone marrow (**C**–**G**), and spleen (**H**–**L**) from *Cish*^+/+^ (WT) and *Cish*^−/−^ (KO) mice injected daily with either PBS (Vehicle) or recombinant mouse granulocyte colony-stimulating factor (G-CSF) as indicated and sacrificed on day 6. Shown are representative images of Giemsa-stained smears (**A**,**C**,**H**) along with corresponding differential counts (**B**,**D**,**I**), frequencies of specific myeloid populations determined using FACS (**E**,**J**), frequencies of relevant clonogenic populations (**F**,**K**), and total cell counts (**G**,**L**). Shown are the mean and SEM, together with statistical significance as determined by two-way ANOVA/Tukey’s multiple comparison test indicated between genotypes (* *p* < 0.05, ** *p* < 0.01, *** *p* < 0.001) or treatments (## *p* < 0.01, ### *p* < 0.001) (n = 3).

**Table 1 ijms-24-12757-t001:** Summary of significant differences in specific myeloid cell populations between *Cish*^+/+^ (WT) and *Cish*^−/−^ (KO) mice.

		KO vs. WT	KO+GM-CSF vs. WT+GM-CSF	KO+G-CSF vs. WT+G-CSF
Blood	Neutrophils	↑	**−**	**−**
Bone marrow	Neutrophils	↑	↑	**−**
	Myeloid DCs	**−**	↑↑	**−**
	CFU-M	**−**	↑↑	**−**
Spleen	Neutrophils	↑↑	↑	**−**
	CFU-G	**−**	↑↑	↑
	CFU-M	**−**	**−**	↑↑

Legend: **−** = no change; ↑ = increase <1.5-fold; ↑↑ = increase >1.5-fold.

## Data Availability

All data analyzed during this study are included in the published article (and associated Appendix A) or are available by request.

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
