# Peer review of "The Role of Cytokine-Inducible SH2 Domain-Containing Protein (CISH) in the Regulation of Basal and Cytokine-Mediated Myelopoiesis"

_ijms, 2023, doi:10.3390/ijms241612757_

Round 1

Reviewer 1 Report

The work by Naser et al. addresses the impact of CISH protein on immune cell frequencies in normal vs. depletion state and in G-CSF and GM-CSF-mediated signaling. The model line of KO mice used in the study was described with KO verification elsewhere (cited by the authors).

The methods employed to describe immune cell subpopulations are relevant and the results are very interesting, however, the manuscript presently contains some confusing issues that require clarification.

First, Figure 1 illustrates the statistically significant differences in

1)    total neutrophil frequency in the KO vs. WT mouse peripheral blood

2)    in myeloid cell frequency in the KO vs. WT mouse spleen and bone marrow, and

3)    in neutrophil frequency in the KO vs. WT mouse bone marrow.

Figures 2 and 3 contain similar columns where no such differences are shown. Presumably, this is due to different numbers of samples analyzed. However, no clear explanation is provided in the text and the data look confusing. Also, could the authors justify the use of a not-so-popular Holm-Sidak method in comparing KO and WT in (Figure 1)??

Second, some data are not shown but referred to in the results and discussion (Lines 69; 114; 213). With no strict limits to word count or Supplementary file size, I would ask the authors to provide the data or omit the references from the text.

Third (minor comment), Lines 78-80 and 126-128 look more like discussion than like results. The authors should consider moving them to Discussion.

Fourth, in Materials and Methods the injection volumes of G-CSF, GM-CSF and PBS are not provided. Also, could the authors justify the use of rhG-CSF?

Fifth (minor comment), the amounts of BM cells and splenocytes used for CFU assays were different. Were the results both recalculated per 10^5 cells as provided in Figure 2 F;K and Figure 3 F;K?

With these issues clarified, the manuscript would be suitable for publication.

Author Response

The work by Naser et al. addresses the impact of CISH protein on immune cell frequencies in normal vs. depletion state and in G-CSF and GM-CSF-mediated signaling. The model line of KO mice used in the study was described with KO verification elsewhere (cited by the authors).

The methods employed to describe immune cell subpopulations are relevant and the results are very interesting, however, the manuscript presently contains some confusing issues that require clarification.

First, Figure 1 illustrates the statistically significant differences in

1)    total neutrophil frequency in the KO vs. WT mouse peripheral blood

2)    in myeloid cell frequency in the KO vs. WT mouse spleen and bone marrow, and

3)    in neutrophil frequency in the KO vs. WT mouse bone marrow.

Figures 2 and 3 contain similar columns where no such differences are shown. Presumably, this is due to different numbers of samples analyzed. However, no clear explanation is provided in the text and the data look confusing. Also, could the authors justify the use of a not-so-popular Holm-Sidak method in comparing KO and WT in (Figure 1)??

> We concur with the Reviewer that the absence of significant basal differences in some of these parameters in Figures 2 and 3 likely relates to the reduced number of samples analyzed (and so reduced statistical power). However, basal myeloid and neutrophil frequencies in the bone marrow in Figure 3 do actually reach significance, with others showing clear trends. Along with additional supporting data from mice on a C57/BL6 background this provides confidence that the differences are genuine. This point is discussed in more detail in the revised Discussion.

> The comparison between KO and WT has been re-analyzed with the alternate Bonferroni-Dunn method that confirmed the differences (revised Figure 1).

Second, some data are not shown but referred to in the results and discussion (Lines 69; 114; 213). With no strict limits to word count or Supplementary file size, I would ask the authors to provide the data or omit the references from the text.

> As requested, the data previously not shown is now included in the revised manuscript or omitted.

Third (minor comment), Lines 78-80 and 126-128 look more like discussion than like results. The authors should consider moving them to Discussion.

> As suggested, these sections of text have been moved to the revised Discussion.

Fourth, in Materials and Methods the injection volumes of G-CSF, GM-CSF and PBS are not provided. Also, could the authors justify the use of rhG-CSF?

> As requested, the volumes of G-CSF, GM-CSF and PBS injected are provided in the revised Section 4.1.

> Human (h) and mouse (m) G-CSF demonstrate complete biological cross-reactivity underpinned by equivalent receptor binding and activity [Nicola 1987 Int J Cell Cloning 5(1):1], with recombinant human G-CSF (rhG-CSF) being the pre-eminent version used for mouse studies in a myriad of contexts (eg. Matsuaki et al. 1996 Int J Immunopharm 18(6-7):363; Campbell et al. 2000 J Leuk Biol 68(1):144; Tsai et al. 2007 J Exp Med 204(6):1273; Hermesh et al. 2012 PLosOne 7(5):e37334; Huttborn et al. Rad Res 191(4):335]. Brief justification for rhG-CSF use is now included in the revised Section 4.1.

Fifth (minor comment), the amounts of BM cells and splenocytes used for CFU assays were different. Were the results both recalculated per 10^5 cells as provided in Figure 2 F;K and Figure 3 F;K?

> In fact, the methodology provided was incorrect. We apologise for this error that is now corrected in the revised Section 4.4. This clarifies that the same number of bone marrow cells and splenocytes (1x10^5) were plated in each case.

Reviewer 2 Report

In this manuscript by Naser et al, the authors describe their results from studies of CISH knockout mice (Cish-/-) in order to identify the role of CISH to regulate myelopoiesis. The hypothesis is that loss of Cish will increase myeloid cell numbers due to its role as a SOCS protein that inhibits signaling pathways activated by Type I cytokines, including GM-CSF and G-CSF. They assessed cell numbers derived from peripheral blood, bone marrow and spleen, plus performed colony forming assays from Cish-/- vs. Cish+/+ mice, either basally or after GM-CSF vs. G-CSF injections. Overall, numbers of cells increase with treatments of GM-CSF or G-CSF in either mutant vs. wild-type animals, as is expected and previously shown. However, their results do show increased numbers of neutrophils in the mutant mice, and differences in numbers of neutrophils and myeloid DCs, plus CFU-M or CFU-G in bone marrow and spleen of the Cish-/- mice vs. Cish+/+, respectively. Together, the data indicate that loss of Cish in mice does appear to cause increased myelopoiesis, but there are some suggestions regarding data presentation as described below. More importantly, the changes in cell numbers are rather subtle (albeit in some cases statistically significant), and there are no mechanistic studies presented that would reveal how loss of CISH affects signaling downstream from GM-CSF, or how loss of Cish affects innate immune functions in the mutant mice. Does loss of Cish cause changes in pathways downstream of GM-CSF, including STAT signaling? Are there any functional defects to the innate immune system with the changes in neutrophils or myeloid DCs in Cish-/- mice? Although the results presented are intriguing and add to our knowledge of how CISH might regulate myelopoiesis, as a descriptive study focused primarily on cell numbers produced in the mutant mice without any in-depth analyses of the mechanisms at play as Cish regulates cytokine-mediated myeloid cell differentiation, or how its loss affects innate immune responses, the impact is diminished.

Introduction: The mechanism of how CISH inhibits cytokine signaling should be described (e.g., interference of the downstream signaling pathways), which should be followed up by analyses of such pathways in cells that lack Cish (either in vivo or use of in vitro models).

 Lines 82-87, the authors mention in the first sentence that GM-CSF injections cause an increase in total neutrophils in the Cish-/- mice, but then in the next sentence state that no differences were observed between Cish+/+ and Cish-/- mice, except with lymphocytes, which contradicts the previous sentence. The data indicate in 2B that total neutrophils in GM-CSF-treated Cish-/- peripheral blood indeed increased, whereas this did not occur with the Cish+/+ mice. Perhaps a rewording of the two sentences would help resolve this reviewer's confusion.

Lines 100-114, this reader becomes bogged down in the descriptions of differences in cell numbers throughout this section as the authors navigate the different tissues or analytical tools being utilized (i.e., peripheral blood vs. spleen, FACS vs. colony-forming assays), therefore a summary graph and/or table that focuses on those cells (e.g. neutrophils vs. DCs vs. CFU-G) that show differences would be greatly help interpretation of the abundant data shown in the graphs.

Section 2.3, as stated for the previous section, a separate set of data/graphs that summarize only the cell types that show significant differences between Cish+/+ vs. Cish-/- mice would be greatly helpful. Most of the data simply show that the cytokines can increase myeloid cell populations, but there are few comparisons that indeed show differences between the genotypes, which is the key data of the manuscript that will support their hypothesis that CISH plays an important role in myelopoiesis. As presented, the impact is lost among the abundant cell types analyzed, most of which show no effect by loss of CISH (and predictable increases observed in both mutant vs. wild-type mice).

Discussion: The authors do mention a possible signaling mechanism that is negatively regulated by CISH, and that the differences in differential numbers produced in response to GM-CSF vs. G-CSF may involve the inability of CISH to dock the G-CSFR, which is good. But this raises the issue again of what molecular differences are observed in the Cish-/- mice during myelopoiesis in the bone marrow (or spleen) vs. Cish+/+ mice, and will such differences affect the overall innate immune responses in the mutant mice. Without this information, it is difficult to judge the impact of CISH on myeloid signaling mechanisms, in particular that activated by GM-CSF vs. G-CSF.

Author Response

In this manuscript by Naser et al, the authors describe their results from studies of CISH knockout mice (Cish-/-) in order to identify the role of CISH to regulate myelopoiesis. The hypothesis is that loss of Cish will increase myeloid cell numbers due to its role as a SOCS protein that inhibits signaling pathways activated by Type I cytokines, including GM-CSF and G-CSF. They assessed cell numbers derived from peripheral blood, bone marrow and spleen, plus performed colony forming assays from Cish-/- vs. Cish+/+ mice, either basally or after GM-CSF vs. G-CSF injections. Overall, numbers of cells increase with treatments of GM-CSF or G-CSF in either mutant vs. wild-type animals, as is expected and previously shown. However, their results do show increased numbers of neutrophils in the mutant mice, and differences in numbers of neutrophils and myeloid DCs, plus CFU-M or CFU-G in bone marrow and spleen of the Cish-/- mice vs. Cish+/+, respectively. Together, the data indicate that loss of Cish in mice does appear to cause increased myelopoiesis, but there are some suggestions regarding data presentation as described below. More importantly, the changes in cell numbers are rather subtle (albeit in some cases statistically significant), and there are no mechanistic studies presented that would reveal how loss of CISH affects signaling downstream from GM-CSF, or how loss of Cish affects innate immune functions in the mutant mice. Does loss of Cish cause changes in pathways downstream of GM-CSF, including STAT signaling? Are there any functional defects to the innate immune system with the changes in neutrophils or myeloid DCs in Cish-/- mice? Although the results presented are intriguing and add to our knowledge of how CISH might regulate myelopoiesis, as a descriptive study focused primarily on cell numbers produced in the mutant mice without any in-depth analyses of the mechanisms at play as Cish regulates cytokine-mediated myeloid cell differentiation, or how its loss affects innate immune responses, the impact is diminished.

> The revised Discussion addresses these issues, as detailed in the specific responses below.

Introduction: The mechanism of how CISH inhibits cytokine signaling should be described (e.g., interference of the downstream signaling pathways), which should be followed up by analyses of such pathways in cells that lack Cish (either in vivo or use of in vitro models).

> The proposed mechanism of how CISH differentially acts between GM-CSFR and G-CSFR signaling is in the revised Discussion. This includes a clearer articulation of the detailed mechanistic studies performed by others in the context of GM-CSFR (esp. Louis et al, 2020), as well as description of additional data in the context of G-CSFR provided in the new Figure S1.

Lines 82-87, the authors mention in the first sentence that GM-CSF injections cause an increase in total neutrophils in the Cish-/- mice, but then in the next sentence state that no differences were observed between Cish+/+ and Cish-/- mice, except with lymphocytes, which contradicts the previous sentence. The data indicate in 2B that total neutrophils in GM-CSF-treated Cish-/- peripheral blood indeed increased, whereas this did not occur with the Cish+/+ mice. Perhaps a rewording of the two sentences would help resolve this reviewer's confusion.

> As suggested, lines 82-87 have been reworded for purposes of clarification.

Lines 100-114, this reader becomes bogged down in the descriptions of differences in cell numbers throughout this section as the authors navigate the different tissues or analytical tools being utilized (i.e., peripheral blood vs. spleen, FACS vs. colony-forming assays), therefore a summary graph and/or table that focuses on those cells (e.g. neutrophils vs. DCs vs. CFU-G) that show differences would be greatly help interpretation of the abundant data shown in the graphs.

> As recommended, a summary table has now been provided of key differences between WT and KO mice (Table 1).

Section 2.3, as stated for the previous section, a separate set of data/graphs that summarize only the cell types that show significant differences between Cish+/+ vs. Cish-/- mice would be greatly helpful. Most of the data simply show that the cytokines can increase myeloid cell populations, but there are few comparisons that indeed show differences between the genotypes, which is the key data of the manuscript that will support their hypothesis that CISH plays an important role in myelopoiesis. As presented, the impact is lost among the abundant cell types analyzed, most of which show no effect by loss of CISH (and predictable increases observed in both mutant vs. wild-type mice).

> As recommended, a summary table has now been provided of key differences between WT and KO mice (Table 1).

Discussion: The authors do mention a possible signaling mechanism that is negatively regulated by CISH, and that the differences in differential numbers produced in response to GM-CSF vs. G-CSF may involve the inability of CISH to dock the G-CSFR, which is good. But this raises the issue again of what molecular differences are observed in the Cish-/- mice during myelopoiesis in the bone marrow (or spleen) vs. Cish+/+ mice, and will such differences affect the overall innate immune responses in the mutant mice. Without this information, it is difficult to judge the impact of CISH on myeloid signaling mechanisms, in particular that activated by GM-CSF vs. G-CSF.

> The Discussion has been revised to provide a better explanation of possible signaling mechanisms.

> The revised Discussion also refers to our on-going studies examining the functional impacts of CISH ablation in the context of immunity to viral and parasitic infections.

Round 2

Reviewer 2 Report

The authors have addressed the concerns of this reviewer, and agree that, based on the data indicating no changes in STAT3 or STAT5 signaling, the next logical step is to assess innate immune responses that go beyond this initial analyses of the Cish-/- mice. The added values of cell numbers plus statistical results in the text are appreciated and important to include. The only criticism would be a minor change to Table 1 to show different "plus" signs to indicate approximate levels of increased numbers of cells in the indicated treatments (e.g., "+" vs. "++", or "+++") rather than simply an upward arrow, which would be more meaningful when comparing the data.

Author Response

> As suggested, Table 1 has been modified to provide an indication of the extent of the disruption in each case.